# A Connectivity Metrics-Based Approach for the Prediction of Stress-Dependent Fracture Permeability

Qinglin Deng [1,2], Xueyi Shang [1,2,*] and Ping He [3]

1   State Key Laboratory of Coal Mine Disaster Dynamics and Control, Chongqing University, Chongqing 400044, China; qinglin@cqu.edu.cn
2   School of Resources and Safety Engineering, Chongqing University, Chongqing 400044, China
3   Chengdu Design Consulting Group, Chengdu 610095, China; hp-amy@foxmail.com
*   Correspondence: shangxueyi@cqu.edu.cn

**Abstract:** Rapid and accurate assessment of fracture permeability is critical for subsurface resource and energy development as well as rock engineering stability. Fracture permeability deviates from the classical cubic law under the effect of roughness, geological stress, as well as mining-induced stress. Conventional laboratory tests and numerical simulations are commonly costly and time-consuming, whereas the use of a connectivity metric based on percolation theory can quickly predict fracture permeability, but with relatively low accuracy. For this reason, we selected two static connectivity metrics with the highest and lowest prediction accuracy in previous studies, respectively, and proposed to revise and use them for fracture permeability estimation, considering the effect of isolated large-aperture regions within the fractures under increasing normal stress. Several hundred fractures with different fractal dimensions and mismatch lengths were numerically generated and deformed, and their permeability was calculated by the local cubic law (LCL). Based on the dataset, the connectivity metrics were counted using the revised approach, and the results show that, regardless of the connectivity metrics, the new model greatly improves the accuracy of permeability prediction compared to the pre-improved model, by at least 8% for different cutoff aperture thresholds.

**Keywords:** fracture permeability; connectivity metrics; numerical modeling; percolation theory

## 1. Introduction

Understanding the flow and transport properties of rock fractures is critical for many underground projects, such as mining engineering, geothermal energy development, shale gas exploitation, and carbon dioxide geological storage. In the case of underground mine engineering, fluids inside the fracture are closely related to ore mining efficiency, rock stability assessment, and groundwater disaster control [1–3]. In addition, for low permeable rock reservoirs, fractures, as the main flow channels for subsurface fluids, determine the efficiency of resource and energy development, as well as the effectiveness of long-term sequestration of hydrocarbons, carbon dioxide, hydrogen energy, etc. [4–6]. One of the key parameters describing the flow and transport properties of rock mass is fracture permeability. Therefore, the estimation of fracture permeability has been a hot research topic [7–11]. The cubic law, as the most commonly used estimation approach, assumes that the fracture surfaces are smooth and parallel [8]. However, rock fractures in nature are composed of rough walls and deformed by stress history, resulting in a decrease in the fracture mean aperture, an increase in the contact area between their upper and lower surfaces, and the degree of tortuosity, thus further deviating from the ideal cubic law.

Over the past decades, many scholars have investigated the effect of heterogeneous geometric surfaces on fracture permeability, mainly including roughness coefficient, contact area, standard deviation of fracture aperture field, and correlation length [7,10,12–15]. On the one hand, the cubic law has been correlated, and the fracture permeability can be directly calculated by incorporating varied geometrical parameters; on the other hand,

quantified scaling functions such as the relationship between fracture normal stiffness and fracture permeability has been proposed, as such the fracture permeability can be indirectly estimated through the normal stiffness, which can be remotely detected with geophysical methods. However, these studies are mainly based on laboratory experiments and numerical simulations, which in general are costly and computationally inefficient, and the results based on a single statistical parameter do not fully reflect the effective permeability of the heterogeneous fracture aperture field [16]. Therefore, some researchers have proposed the use of percolation theory to estimate the permeability of heterogeneous porous media. For instance, different connectivity regimes were determined for three-dimensional natural fault networks assuming a power law length distribution [17], and the reservoir permeability and connectivity were evaluated by percolation theory [18]. Compared with the conventional reservoir modeling approach, the computational efficiency of the percolation theory method was greatly improved. Moreover, this method is closely related to fracture geometry. For example, the percolation through self-affine surfaces was found to be controlled by the largest wavelength of the height distribution [19]; a power law relationship between the heat conductivity and the statistical parameters of the rough fractures was proposed based on the concept of the percolating cluster [20], and the fracture permeability was shown as a function of the percolation probability and the fracture contact area [21]. In addition, the percolation threshold of a three-dimensional binary random field was shown to be associated with the correlation length and the finite size of the field [22].

The percolation theory divides the study area into permeable and non-permeable zones, assuming that flow depends on the connectivity of fracture flow paths, which can be quantitatively described by connectivity metrics. Knudby et al. measured the connectivity of two-dimensional binary random fields and found that the information on connectivity can significantly improve the hydraulic behavior close to the percolation threshold [23]. By studying the relationship between connectivity and percolation theory, Hovadik and Larue defined the types of connectivity of reservoirs, the proposed methods for the measurement of connectivity, and identified the controlling factors of reservoir connectivity [24]. Tyukhova and Willmann showed that based on information on the resistance and geometry of the connected channel network, the static connectivity metrics can predict effective flow and transport of heterogeneous fields through comparison with flow simulations [25]. In previous studies, the percolation thresholds on self-affine surfaces were investigated [26], and the difference between the static and dynamic connectivity metrics for the characterization of heterogeneous porous reservoirs was clarified [27]. In detail, the static connectivity metrics are only related to the connectivity geometrical parameters, such as hydraulic conductivity or geological structures. In comparison, the dynamic connectivity metrics are time-dependent and rely on physical processes, like fluid flow and transport. The two types of connectivity metrics can be linked; however, the relations are extremely complex. With the above research basis, Javarmand et al. [16] recently successfully applied static connectivity metrics for the first time based on percolation theory to estimate the permeability of rough deformed fractures. For three commonly used connectivity metrics, the accuracy of the permeability prediction was 72%, 55%, and 63%, respectively, which needs to be further improved.

To this end, a new approach by revising the connectivity metrics was proposed in this study to better characterize the flow properties of the deformed fracture. The idea lies in the fact that there exist trapped fluid regions within the fracture, especially for fractures under high normal stress [15,16]. In this case, the connectivity of the fracture must be rechecked to eliminate or reduce the effects of large isolated aperture areas. For this purpose, we first numerically generated fractures characterized by different geometrical statistical parameters (mainly considering the fractal dimension and the mismatch length), and then obtained the deformed fracture aperture field by progressively applying normal stresses. Furthermore, by simulating the fracture flow at low Reynolds numbers, we obtained a fracture permeability dataset. We then analyzed the fracture aperture percolation threshold, calculated the connectivity metrics of all generated fractures, and corrected them according

to the potential trapped fluid regions. Finally, we compared the permeability–connectivity metric models before and after correction.

## 2. Materials and Methods

### 2.1. Fracture Aperture Generation

Synthetic fractures were used in this study to generate a large number of fractures with controllable surface properties. These were implemented by using *SynFrac,* which creates two opposing rough fracture surfaces that can well approximate natural rock surfaces by taking into account their statistical parameters, roughness, matchedness, and anisotropy [28–30]. The fracture size is $100 \times 100$ mm and discretized by $512 \times 512$ uniform square grid cells in the $xy$-plane. This resolution is lower than the one of acceptable error [16]. The standard deviation (0.1 mm) remains the same for all fractures, and no anisotropy was considered. To characterize varying fracture roughness, the fractal dimension was set to 2.1–2.5, corresponding to roughness exponent 0.5 to 0.9 for natural rock surfaces, as commonly observed both in the laboratory and field [31–34]. In *SynFrac*, the generated top and bottom surfaces of the fracture will be adjusted until a single contact point is left [35]. To obtain the same mean aperture for all generated fractures, the fracture aperture was manually regulated to 0.35 mm. Moreover, by changing the seeds of the random number generator, namely the Park and Miller pseudo-random number generator, a series of fractures with the same properties can be created [36].

In addition, the surface matching is crucial to fracture characterization. Previous studies showed that the geometry of the two opposing fracture surfaces is correlated at long wavelengths but uncorrelated at short wavelengths in most cases [37–40]. Thus, the length over which the fracture surfaces are correlated is called the mismatch length, or correlation length. Several models were implemented in *SynFrac* to describe the mismatch length, here the Brown model was used in our study [41] to set the mismatch length to 5 mm, 10 mm, and 20 mm, referring to commonly observed values [37,38,42,43].

For each generated aperture field, 10 normal stress levels were stepwise applied to simulate fracture closure under normal stress. The numerical simulations of fracture surface contact were solved by a fast Fourier transform (FFT)-based convolution approach integrated with the boundary element method (BEM) [44]. In this study, only elastic deformation was considered since the target is to obtain deformed aperture fields rather than a real fracture closure process, which may be better described by elastoplastic deformation [11,45,46]. In terms of linear elasticity, the normal closure $u(x, y)$ under normal stress $\sigma(x, y)$ is calculated as follows:

$$u(x, y) = G(x, y) * \sigma(x, y) \tag{1}$$

where $G(x, y)$ is the Green's function, and the symbol $*$ represents the convolution.

For non-period fracture aperture field in this study, the Green's function can be written as follows:

$$G(x, y) = M/\pi/E^* \tag{2}$$

where $M$ is a geometrical parameter, which is related to the coordinates of the calculated point and the measured resolution of the aperture field; $E^*$ is the effective elastic modulus, given by:

$$E^* = \frac{1}{2(1 - \nu^2)/E} \tag{3}$$

In this study, the elastic modulus $E$ and Poisson ratio $\nu$ were assumed to be 60 GPa and 0.25, corresponding to typical mechanical properties of granite [10,11,47]. The initial normal loading was set to an extremely low value to allow the fracture surfaces to come into contact (contact area below 0.1%). For all the stress levels, the contact area varies between 0 and 50% depending mainly on the fractal dimension, forming different flow channels and, therefore, the connectivity of the aperture fields.

In total, 450 aperture fields were generated for the purpose of statistics by considering 10 stress levels, 3 fractal dimensions, 3 mismatch lengths, and 5 seeds for each synthetic fracture. As an example, Figure 1 shows fracture aperture distributions at several normal stress levels for a fractal dimension of 2.5 and a mismatch length of 20 mm.

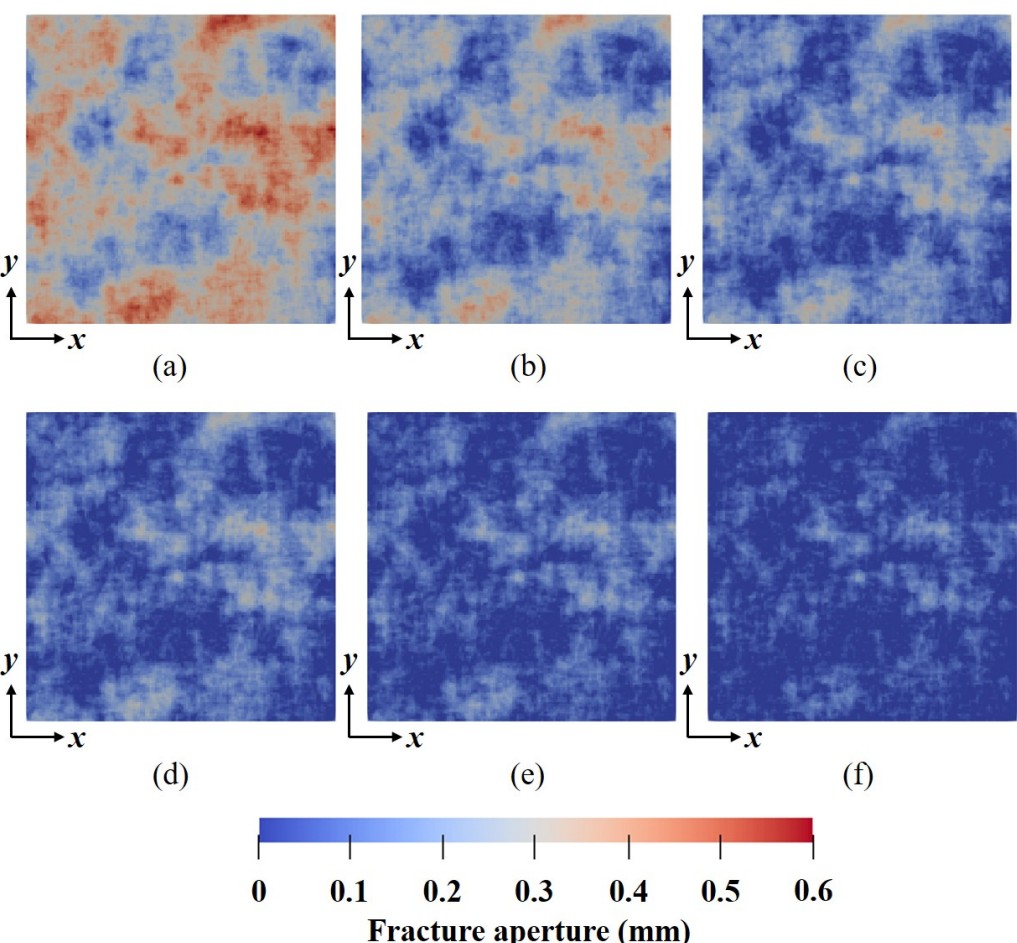

**Figure 1.** Fracture aperture distributions under increasing normal stress. (**a**) 0.001 MPa; (**b**) 1 MPa; (**c**) 5 MPa; (**d**) 10 MPa; (**e**) 20 MPa; (**f**) 40 MPa.

### 2.2. Fluid Flow along Fractures

Assuming single phase, fully saturated, steady-state Newtonian fluid flow along the rock fracture, the governing equation is simplified to the Reynolds equation (i.e., local cubic law, LCL) from the Navier–Stokes (NS) equation, which can be written as follows:

$$\nabla \cdot \left[ \frac{d(x,y)^3}{12\mu} \nabla P \right] = 0 \tag{4}$$

where $d$ is fracture local aperture, $\mu$ is fluid viscosity, and $P$ is fluid pressure.

In this study, Equation (4) was solved by the open-source software MOOSE/Golem (https://github.com/topics/moose-framework accessed on 5 February 2024) [48], developed based on the finite element method (FEM). For boundary conditions, constant pressure boundaries were applied to the inlet and outlet sides of the fracture (marked in blue in Figure 2), and no-flow boundaries were imposed on the other sides (marked in red in Figure 2), which forces the fluid flows in the $x$-direction in Figure 2. The simulation started with a zero-pressure field. To ensure a low Reynolds number, a pressure drop of 10 Pa was

set along the flow direction. Once the pressure field and velocity field inside the fracture were obtained, the hydraulic aperture $d_h$ can be obtained by applying Darcy's law [49,50]:

$$d_h = \left[ \frac{12\mu\dot{V}}{(\nabla P/L)w} \right]^{1/3} \tag{5}$$

where $\dot{V}$ is the total flow rate, and $L$ and $w$ are fracture lengths along and perpendicular to the flow direction, respectively.

Finally, the fracture permeability $k$ is calculated as follows:

$$k = d_h^2/12 \tag{6}$$

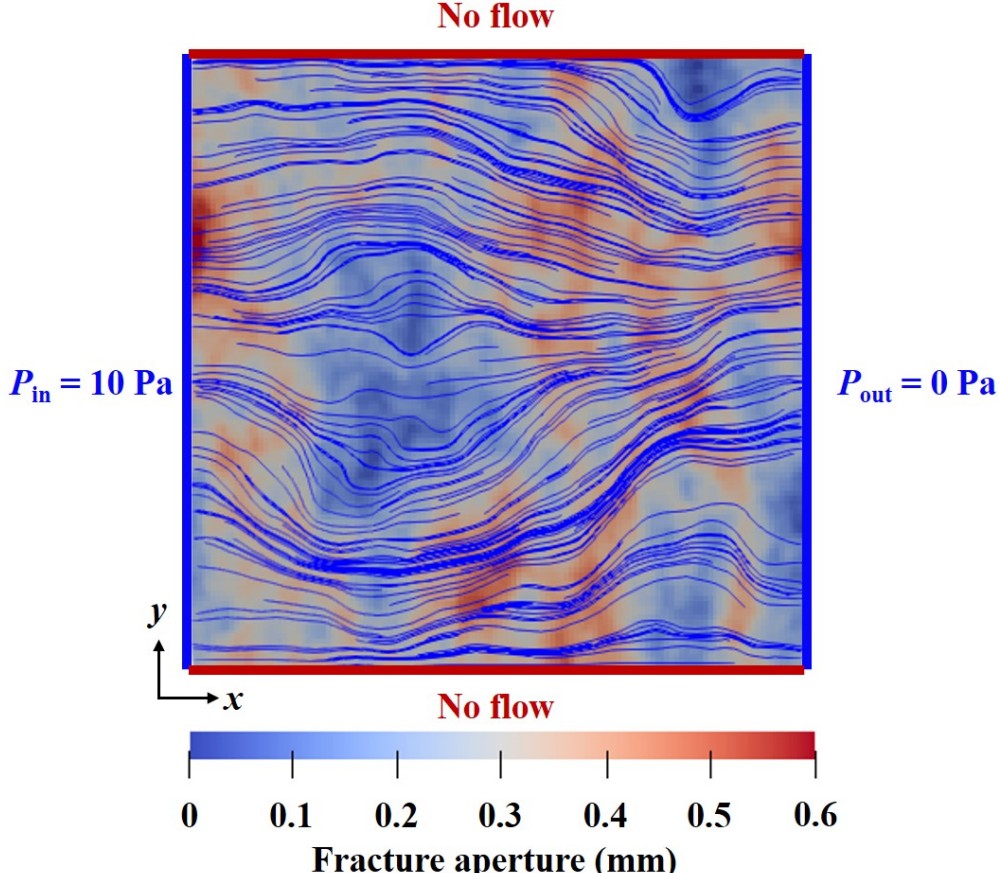

**Figure 2.** Flow boundary conditions and aperture distribution of a generated fracture, the blue curve indicates the flow traces.

*2.3. Percolation Theory and Connectivity Metrics*

To describe the connectivity, one of the indicators that can be used is the percolation theory, which mainly deals with connectivity on an infinite Bernoulli lattice [27]. The percolation theory determines the evolution of the shape and size of clusters as the probability of site occupancy $p$. There exists a proportion $p_c$, If $p > p_c$, a unique cluster with infinite volume occurs with a probability equal to 1. If $p < p_c$, this occurs with probability 0. Such $p_c$ is called the percolation threshold, where $p_c \neq 0$ and $p_c \neq 1$ [51]. Figure 3a,b depict the concept of percolation theory by using a four-neighbor and eight-neighbor algorithm, respectively. The element $A$ connects with 4 elements (marked in blue) through its edges for the former case while it connects 7 elements (marked in blue) through both its edges and nodes for the latter case. Thus, for the same network, 13 clusters were formed when applying the four-neighbor algorithm (Figure 3c), whereas only 5 clusters were determined

in terms of the eight-neighbor algorithm (Figure 3d), and a connected path was also created (cluster 3, from left to right). In this study, the eight-neighbor algorithm was selected to determine the accumulated cluster of the fracture aperture field as it mimics more realistic flow patterns [16]. Moreover, as can be seen from Figure 3c,d, the percolation theory divides the heterogeneous aperture field into a binary field, i.e., a permeable phase and an impermeable phase. This requires a cutoff aperture threshold in practical application, and this threshold is the percolation threshold that regulates whether the fracture is open or closed to flow.

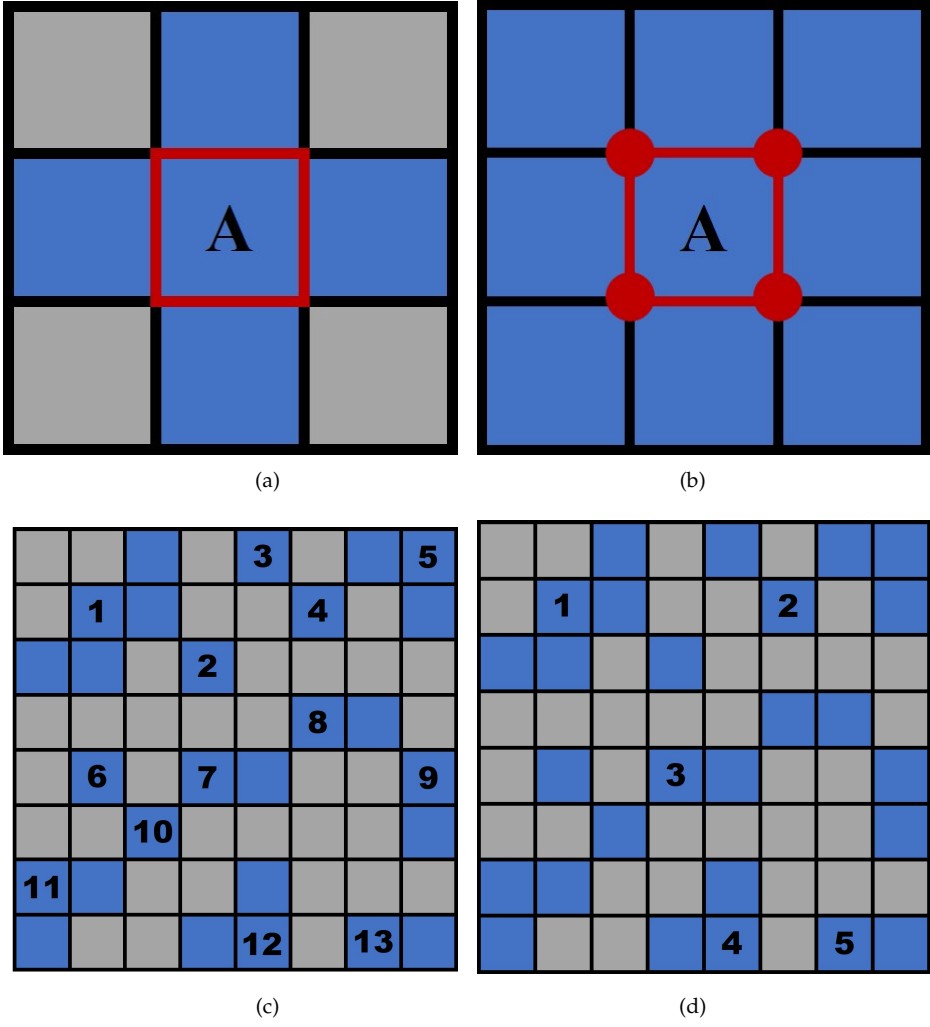

**Figure 3.** Cluster identification for element A using the (**a**) 4-neighbor algorithm (red box) and (**b**) 8-neighbor algorithm (red box with four dots); (**c**) 13 clusters were identified by using the 4-neighbor algorithm for a network; (**d**) 5 clusters were identified by using the 8-neighbor algorithm for the same network.

To obtain the connectivity or permeability of the aperture field above the percolation threshold, Javanmard et al. proposed three static connectivity metrics [16]. Here, we selected two of these metrics that have the highest and lowest probability of predicting fracture permeability, which were briefly described below.

The first connectivity metric $\Theta(z)$ is defined as the probability that two sites belong to the largest cluster. When $p \leq p_c$, $\Theta(z) = 0$; when $p > p_c$, $0 < \Theta(z) < 1$. This indicates that this connectivity metric will be close to zero for all clusters that are sufficiently small and will be one for the entire field in the permeable phase.

The second connectivity metric $\Gamma(z)$ is defined as the probability that a pair of sites belong to the same cluster. Similar to the connectivity metric $\Theta(z)$, $\Gamma(z)$ will approach

zero when clusters are small, and it will be one when the whole permeable phase falls into the same cluster. It can be calculated as the ratio of the number of connected pairs to the number of all pairs in the permeable phase as follows:

$$\Gamma(z) = \frac{1}{n_p^2} \sum_{i=1}^{N} n_i^2 \tag{7}$$

where $n_p$ is the number of sites in the permeable phase, $N$ is the number of clusters, and $n_i$ is the number of sites in cluster $i$.

The computation of connectivity metrics is based on cluster identification [27]. For $\Theta$, only the largest cluster was considered, and for $\Gamma$, smaller and non-percolating clusters were also included. However, isolated or unconnected aperture regions of the fracture may contribute little or nothing to the entire flow field, as indicated by clusters 1, 2, 4, and 5 in Figure 3d. Therefore, considering these regions in the calculation of connectivity metrics may bring relatively large errors in describing the physical processes of the fluid flow along the fracture, in particular when the mean aperture of these regions is large [16]. Therefore, a coefficient, $f$, was introduced here to revise the connectivity metrics, and the revised connectivity metrics can be written as follows:

$$\Theta' = f\Theta \tag{8}$$

$$\Gamma' = f\Gamma \tag{9}$$

The coefficient, $f$, reflects the influence of the trapped regions of a large aperture. When searching for the clusters, the mean aperture of these trapped regions will be checked, and for any value larger than a given threshold, i.e., twice the mean aperture of the entire fracture field in this study, the clusters will be redetermined depending on the number of clusters with large apertures, and accordingly, the connectivity metrics will be recalculated.

## 3. Results and Discussions

### 3.1. Fracture Permeability under Normal Stress from Numerical Simulations

Figure 4 shows the stress-dependent fracture permeability as a function of the mechanical aperture in a log-log plot for the fracture dataset. In general, the numerically obtained permeability (dark blue balls) deviates from the cubic law (dark line), this is manifested by the fact that the calculated fracture permeability is all below the one predicted by the cubic law at the same mechanical apertures. This is consistent with many previous studies [8,15,29,49,52].

In addition, we compared several established models to our numerical data and found that the upper and lower limits can be roughly described by models proposed by [53] (orange line) and [8] (red line), respectively, as follows:

$$d_h = d_m \left(1 - \frac{d_m^\sigma}{d_m}\right)^{1/3} \tag{10}$$

$$d_h = \frac{d_m}{\left(1 - 1.5\left(\frac{d_m^\sigma}{d_m}\right)^2\right)^{1/3}} (1 - 2A_c) \tag{11}$$

where $d_m^\sigma$ is the standard deviation of the aperture field, and $A_c$ is the contact area.

The above models used the parameter $d_m^\sigma/d_m$. In between, the relationships between fracture permeability and mechanical aperture were given by many authors using varying parameters. For instance, ref. [54] also used the ratio $d_m^\sigma/d_m$ for the estimation of fracture permeability (yellow line); Ref. [55] connected the cubic of the hydraulic and mechanical apertures by incorporating fracture contact area (cyan line), which was complemented by [7] (dark green line) by considering the aspect ratio of the ellipse; Ref. [56] (pink line)

related the $d_h$ to $d_m$ using the joint roughness coefficient (JRC), indicating the effect of roughness on the $d_h - d_m$ relationship; similarly, ref. [57] (purple line) proposed a modified cubic law by introducing a roughness factor. The predicting accuracy of each method is presented in Figure 4, ranging from 75% to 88%. Note that these values are averaged and the curves shown in Figure 4 are compared with only one identical fracture. Although each study contributes to the permeability–aperture model from different perspectives and each model possesses its advantages, they were all obtained based on specific conditions and, therefore, all have certain limitations. Currently, there is no single model that can completely describe the relationship between fracture permeability and mechanical aperture and, thus, further exploration, such as advancing the prediction approach of the fracture permeability, is required.

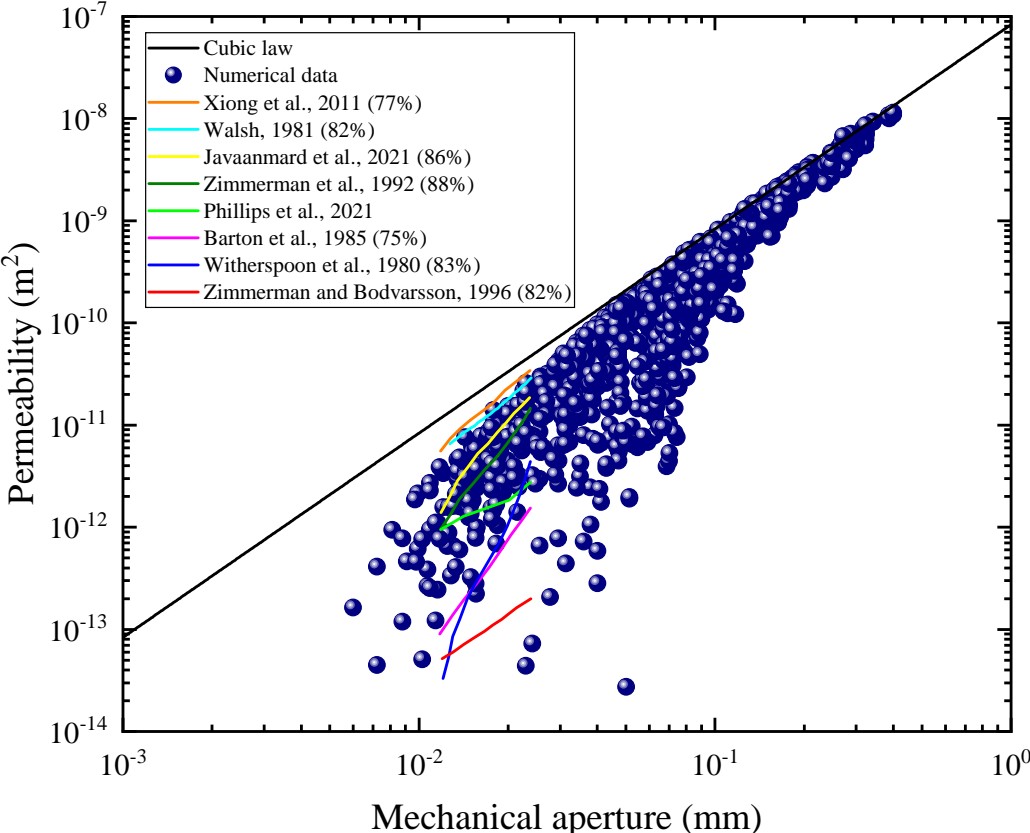

**Figure 4.** Fracture permeability obtained from the numerical simulations (blue balls) as a function of mechanical aperture. The data was compared with experimental data ([29]) and several previous derived models ([7,8,53–57]).

*3.2. Percolation Threshold of the Fracture Aperture Field*

The percolation threshold was determined in terms of the probability of the permeable phase, it is the threshold at which no percolating clusters exist just before a given cutoff aperture threshold. Figure 5a,b show the averaged percolation threshold probability as a function of the mismatch length and the fractal dimension under increasing normal load, respectively. In general, the percolation threshold increases with increasing normal load and decreasing fractal dimension. The results agree with previous studies [16,20], which can be interpreted as the fact that lower normal loading and a higher fractal dimension generally correspond to a lower fracture contact area and, therefore, a lower aperture percolation threshold [20,58]. However, there is no clear relationship between the percolation threshold and mismatched length, this is due to the fact that the mismatch length only regulates the scale of the contact and open regions of the fracture and is not directly related to the topology of the fracture [40,58].

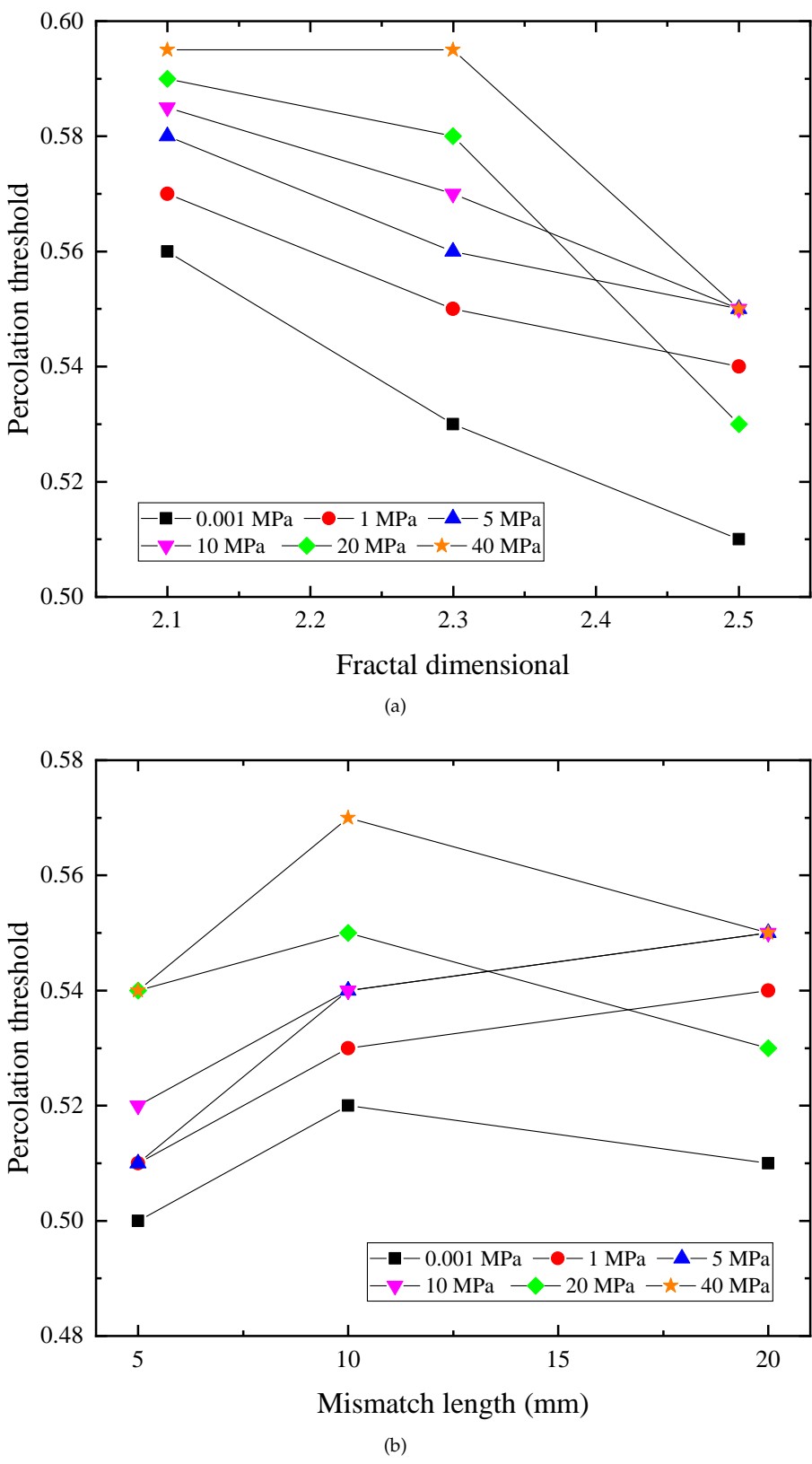

(a)

(b)

**Figure 5.** Percolation threshold of the fracture aperture field as a function of (**a**) mismatch length and (**b**) fractal dimension under increasing normal load.

### 3.3. Fracture Permeability and Connectivity Metrics

Based on the percolation theory, previous studies (e.g., [16,25]) proposed a power law relationship between the fracture permeability and the connectivity metric as follows:

$$k = aX^b \tag{12}$$

where $X$ represents the connectivity metric (i.e., $\Theta$ and $\Gamma$ in this study), and $a$, $b$ are constants.

For the cutoff thresholds $d_c$ of 0, 10, 20, and 30 μm, we compared the fitting parameters $a$, $b$ and $R^2$ from Javanmard et al. [16] and our study, as detailed in Table 1. All the fitted curves are quite close, with the coefficient of determination $R^2$ around 75% for $\Theta$ and 55% for $\Gamma$, both of which are a few percentage points higher than the accuracy of the fit obtained from Javanmard et al. [16]. For both connectivity metrics, the larger the cutoff threshold, the higher the fitting accuracy. This is because a larger $d_c$ can separate the aperture field into well-bounded contact and non-contact areas, so that some regions may not commit to the overall flow, e.g., large open regions [15,54,59], or by being trapped [15,49,60]. Moreover, a non-zero $d_c$ provides a more accurate description of the real fracture flow, though the larger $d_c$ should be avoided since it may misidentify the percolating zone as a non-percolating zone within the fracture [16].

**Table 1.** Comparison of the fitting parameters using Equation (**??**) between data from [**?** ] (D1), our data from old metrics (D2), and our data from new metrics (D3).

| Data | $d_c$ (μm) | $\Theta$ | | | $\Gamma$ | | |
|------|-----------|----------|------|-------|----------|--------|-------|
| | | a | b | $R^2$ | a | b | $R^2$ |
| D1 | | $10 \times 10^{-10}$ | 25.47 | 0.69 | $3 \times 10^{-10}$ | 220.53 | 0.45 |
| D2 | 0 | $1 \times 10^{-9.38}$ | 26.21 | 0.71 | $1 \times 10^{-9.18}$ | 210.57 | 0.50 |
| D3 | | $1 \times 10^{-9.29}$ | 27.15 | 0.79 | $1 \times 10^{-9.27}$ | 206.88 | 0.62 |
| D1 | | $7 \times 10^{-10}$ | 7.37 | 0.72 | $2 \times 10^{-10}$ | 8.32 | 0.36 |
| D2 | 10 | $1 \times 10^{-9.26}$ | 6.71 | 0.74 | $1 \times 10^{-9.35}$ | 8.09 | 0.51 |
| D3 | | $1 \times 10^{-9.50}$ | 7.22 | 0.82 | $1 \times 10^{-9.52}$ | 8.07 | 0.60 |
| D1 | | $6 \times 10^{-10}$ | 3.70 | 0.70 | $3 \times 10^{-10}$ | 3.69 | 0.48 |
| D2 | 20 | $1 \times 10^{-9.21}$ | 3.55 | 0.75 | $1 \times 10^{-9.34}$ | 3.76 | 0.55 |
| D3 | | $1 \times 10^{-9.33}$ | 3.63 | 0.84 | $1 \times 10^{-9.02}$ | 3.89 | 0.67 |
| D1 | | $7 \times 10^{-10}$ | 2.55 | 0.72 | $3 \times 10^{-10}$ | 2.65 | 0.55 |
| D2 | 30 | $1 \times 10^{-8.88}$ | 2.75 | 0.77 | $1 \times 10^{-9.14}$ | 2.63 | 0.58 |
| D3 | | $1 \times 10^{-8.91}$ | 2.86 | 0.87 | $1 \times 10^{-8.97}$ | 3.15 | 0.72 |

For a better comparison, Figure 6a,b show the numerically estimated fracture permeability as a function of the connectivity metric $\Theta$ and $\Gamma$ at the cutoff threshold of 30 μm, respectively. The fitting lines using Equation (12) were marked in red and compared with the results obtained from [16] (dark line). For the two connectivity metrics, we obtained the power law functions to estimate fracture permeability as follows:

$$k = 1 \times 10^{-8.88}\Theta^{2.75} \tag{13}$$

$$k = 1 \times 10^{-9.14}\Gamma^{2.63} \tag{14}$$

Equations (13) and (14) both yield quite close coefficients of determination $R^2$ as Javarmand et al., i.e., 0.77 vs. 0.72 for $\Theta$, and 0.58 vs. 0.55 for $\Gamma$. Similarly, the connectivity metric $\Theta$ shows a higher accuracy of permeability prediction compared to $\Gamma$.

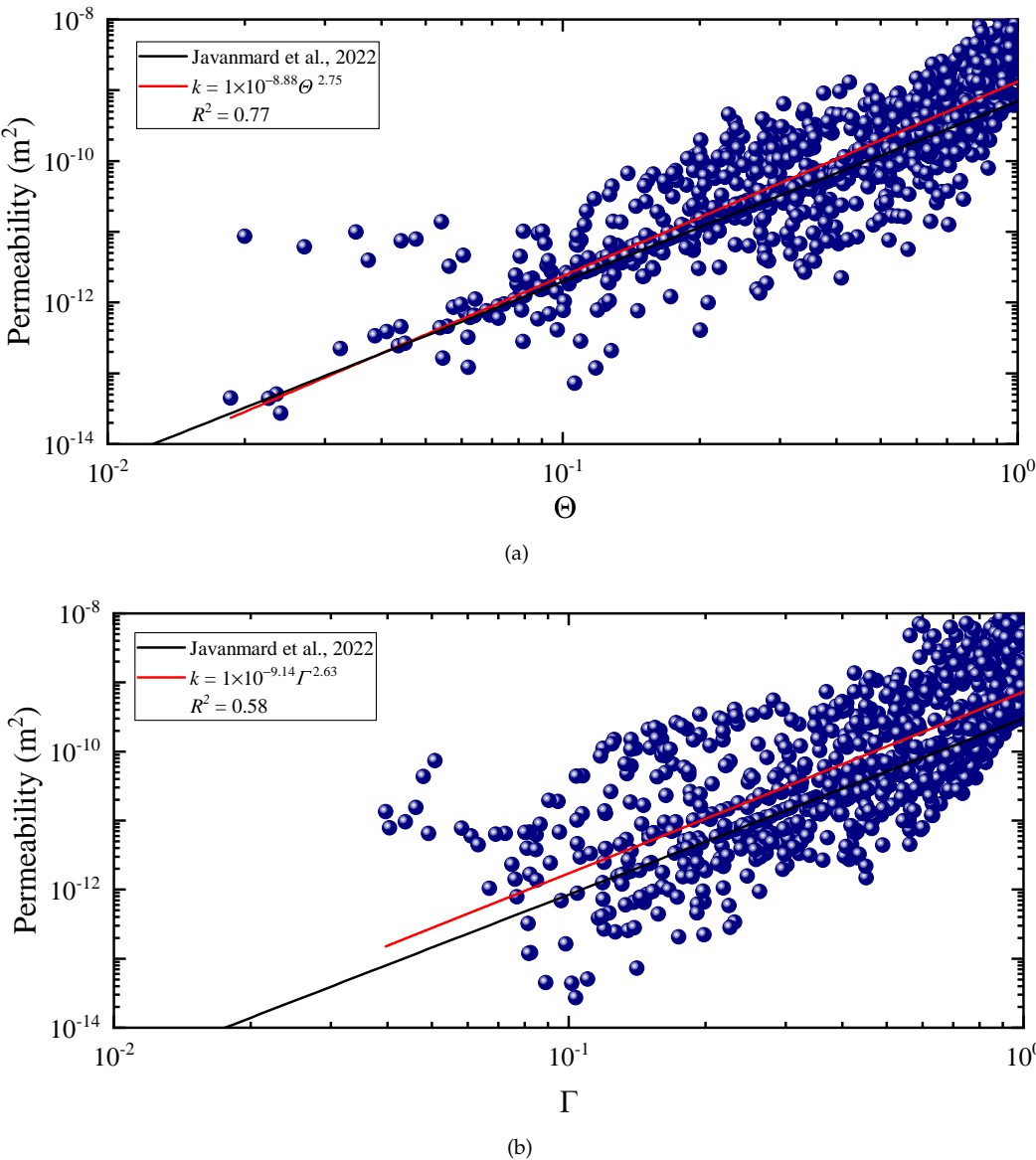

**Figure 6.** Fracture permeability as a function of connectivity metrics (**a**) $\Theta$ and (**b**) $\Gamma$; the results derived from this study and [16] were indicated by red and black lines, respectively.

In this study, we propose to revise the two connectivity metrics as $\Theta'(z)$ and $\Gamma'(z)$ to better characterize the flow properties of the deformed fracture. Figure 7a,b show fracture permeability as a function of the revised connectivity $\Theta'(z)$ and $\Gamma'(z)$, respectively. Using the fitting model Equation (12), we obtain the following relationship between fracture permeability and connectivity metrics for the cutoff threshold of 30 μm:

$$k = 1 \times 10^{-8.91}\Theta'^{2.86} \tag{15}$$

$$k = 1 \times 10^{-8.97}\Gamma'^{3.15} \tag{16}$$

As shown in Figure 7, we then compare the new models (red lines) with the older models (Equations (13) and (14), dark lines). There are only small differences between the predicted fracture permeability, with the largest difference being less than half of an order of magnitude over the range of connectivity metrics studied. However, all data are more centralized towards the predictive model under the new connectivity metrics. This results in a higher coefficient of determination $R^2$. For $\Theta'$ and $\Gamma'$, the fitting accuracy improves by

10 percentage points (from 77% to 87%) as well as 14 percentage points (from 58% to 72%), respectively. For other cutoff thresholds, the prediction accuracy has also been improved by at least 8 percentage points for the two metrics, as given in Table **??**, demonstrating the substantial advantages of our new prediction model.

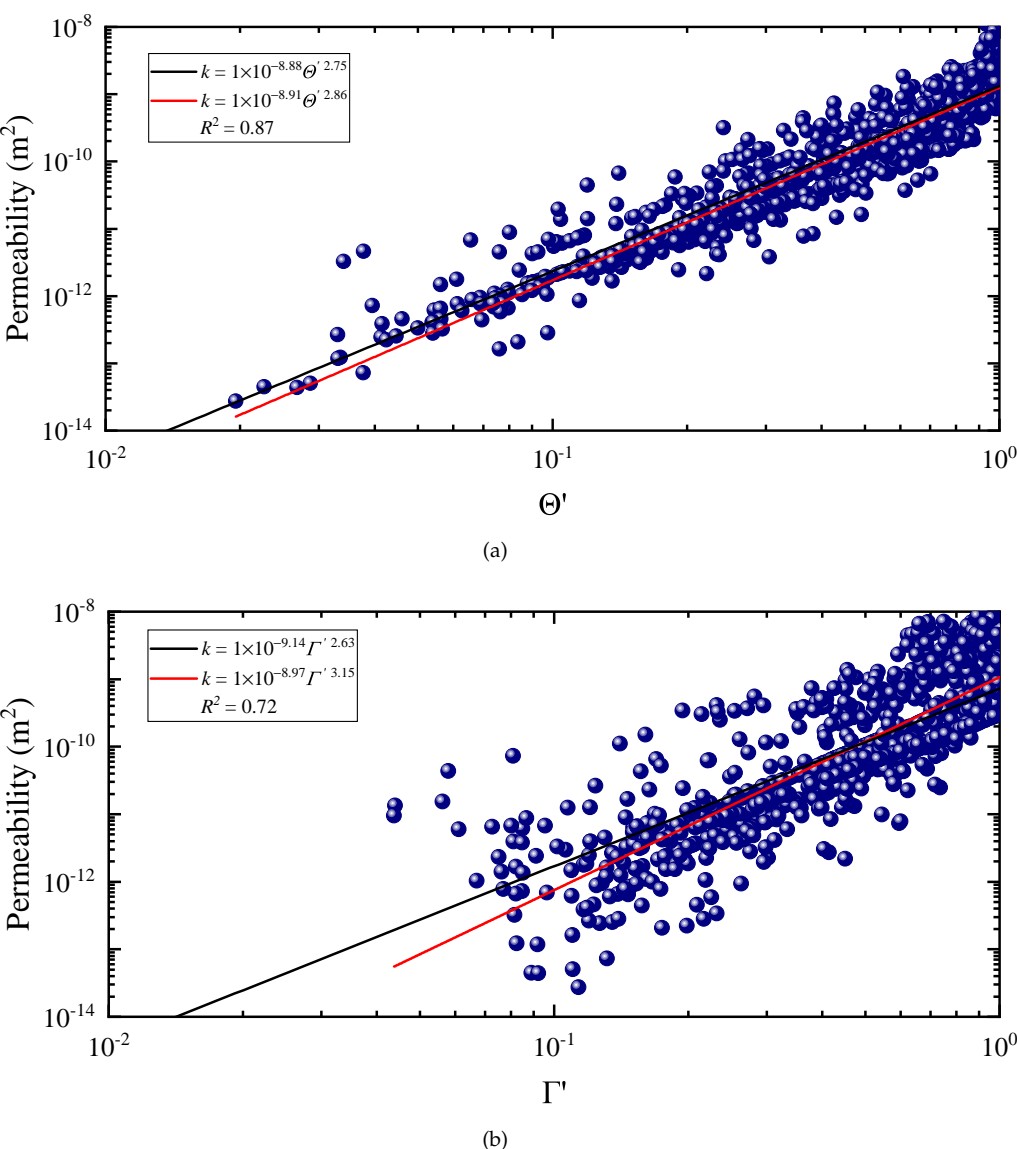

(a)

(b)

**Figure 7.** Fracture permeability as a function of revised connectivity metrics (**a**) $\Theta'$ and (**b**) $\Gamma'$; the results based on the revised connectivity metrics and the common connectivity metrics were indicated by red and dark lines, respectively.

Although both the connectivity metrics have been improved, and the connectivity metric $\Gamma'$ grows to a greater extent than the connectivity metric $\Theta'$ (with an average growth of 12% vs. 8%), $\Theta'$ still has a higher permeability prediction accuracy than that of the $\Gamma'$ (averagely about 18%). Notably, the fracture permeability prediction accuracy by the connectivity metric $\Theta'$ reaches more than 80%, which is comparable with most common permeability models obtained with laboratory experiments and numerical modeling ([7,8,29,53–57], see Figure 4). Moreover, the information required in this study is no more than that of common numerical modeling, yet presents fast and inexpensive computation [27]. In the future, three-dimensional or dynamic connectivity metrics can be identified for application to complex flow and transport processes.

## 4. Conclusions

In this study, we improved two common connectivity metrics, $\Theta$ and $\Gamma$, mainly by introducing the negative impact of isolated/trapped large aperture regions within fractures. A database of fracture permeability was constructed by simulating flow along fractures subjected to stepwise increasing normal stresses. Based on the above work, we evaluated and compared the accuracy of fracture permeability prediction using improved and non-improved connectivity metrics. The main conclusions are summarized as follows:

(1) With the increase in normal loading, the fracture permeability deviates from the cubic law to an increasing degree. For the generated fracture dataset in this study, common fracture permeability prediction models based on fracture geometrical parameters are established by laboratory experiments and numerical simulations with an accuracy of 75% to 88%;

(2) The flow percolation threshold is affected by the fractal dimension of the fracture and the stress variations, independent of the mismatch length, and the connectivity metrics of the fracture aperture field based on the percolation theory can be quickly estimated, which is related to the set cutoff aperture threshold;

(3) By fitting a power law model to permeability–connectivity metrics, similar permeability prediction results and accuracy to the previous study are obtained. Using the improved connectivity metrics, the permeability prediction accuracy is improved by 8 to 15 percentage points for different cutoff aperture thresholds, with a maximum fitting accuracy of 0.87. In particular, for the connectivity metric $\Theta'$, the accuracy remains almost over 80%, comparable with common previous fracture permeability prediction models; however, the computation of our approach can be easily achieved, showing an advantage over previous methods.

**Author Contributions:** Conceptualization, Q.D. and X.S.; methodology, Q.D.; software, Q.D.; validation, X.S. and P.H.; formal analysis, X.S.; investigation, P.H.; resources, Q.D.; data curation, Q.D. and P.H.; writing—original draft preparation, Q.D.; writing—review and editing, X.S.; visualization, Q.D. and P.H.; funding acquisition, Q.D. All authors have read and agreed to the published version of the manuscript.

**Funding:** This research was funded by the China Postdoctoral Science Foundation (2023M740385), the Fundamental Research Funds for the Central Universities (2023CDJXY-006), and the Postdoctoral Fellowship Program of CPSF (GZC20233326).

**Data Availability Statement:** The data that support the findings of this study are available from the corresponding author upon request.

**Acknowledgments:** The authors would like to thank the High-Pressure Water Jet Research Group of Chongqing University for their generous support.

**Conflicts of Interest:** Author Ping He was employed by the company Chengdu Design Consulting Group. The remaining authors declare that the research was conducted in the absence of any commercial or financial relationships that could be construed as a potential conflict of interest. The authors declare that they have no known competing financial interests or personal relationships that could have appeared to influence the work reported in this paper.

## Abbreviations

The following abbreviations are used in this manuscript:

| | |
|---|---|
| LCL | local cubic law |
| FFT | fast Fourier transform |
| BEM | boundary element method |
| NS | Navier–Stokes |
| FEM | finite element method |
| JRC | joint roughness coefficient |

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
