# Peer review of "A Connectivity Metrics-Based Approach for the Prediction of Stress-Dependent Fracture Permeability"

_water, doi:10.3390/w16050697_

Round 1

Reviewer 1 Report

Comments and Suggestions for Authors

The manuscript is devoted to developing a methodology of assessment of fracture permeability of rock mass under stress. Authors used numerically generated synthetic fractures with controllable surface properties, the cubic law  and percolation theory for creating a new approach of predicting stress-dependent fracture permeability. The findings of research improve the accuracy of fracture permeability prediction.

MINOR COMMENTS

1) The authors state that “Fracture permeability deviates from the classical cubic law under the effect of roughness as well as geological stress.” But in case of ore or sedimentary rock mining, the stress state of rock mass may be sufficiently changed as a result of the influence of mining-induced stress (for example,  bearing pressure zones -  areas affected by mining operations (around the large cavities) where stress may be increased 2-5 times relatively geostatic level of stress). This fact should be taken into account if you consider subsurface resources and rock engineering stability. It would be better not to mention only “geological stress” or add “mining-induced stress”.   

2) Some conclusions 0f the manuscript seem obvious:  “With the increase of normal loading, the fracture permeability deviates from the cubic law to an increasing degree, and all previous permeability and mean aperture models based on fracture geometrical parameters established by laboratory experiments and numerical simulations were only partially effective in predicting the permeability and were typically inefficient and costly.

It is known that the cubic law has an exact result only for a fracture bounded by two smooth, parallel plates. Real fractured rock mass is very little like such plates. As a result, there are a lot of methods for improving the accuracy of the cubic law. To my mind, the authors of the manuscript propose just one more corrected method of permeability predicting whose accuracy should be compared with other existing methods.

Reviewer 2 Report

Comments and Suggestions for Authors

Review of manuscript “A Connectivity Metrics Based Approach for the Prediction of Stress-dependent Fracture Permeability”, by Qinglin Deng, Xueyi Shang, and Ping He [Paper WATER 2886607]. Manuscript received by reviewer on 15 February 2024; review submitted on 23 February 2024. 

This paper develops a new approach to modeling stress-dependent fracture permeability, based on percolation theory, and specifically using the concept of connectivity metrics. 

The Synfrac software was used to creates two opposing 100×100 mm rough fracture surfaces with specific roughness and fractal dimension. The opposing surfaces were then mathematically compressed, to create different levels of contact area. The Reynolds lubrication equation, solved using the finite element method, was then used to simulate viscous flow through the fracture. Several connectivity metrics were then used to estimate the fracture permeability, and were found to improve the accuracy of the permeability estimates. 

This is an interesting paper, and probably deserves to eventually be published. I have only two comments/suggestions, which the authors should address before the paper could be accepted.

1. The authors have quantified the ability of the fracture to transmit fluids using the “permeability”. But I think that their equation (5) is incorrect, as I will try to show, using the notation of the present paper. Darcy’s law has the form V = kAΔP/μL, where V is the volumetric flowrate in units of m3/s, L is the length of the specimen in the direction of flow, and A is the cross-sectional area normal to the flow, and k is the permeability with units of m2. We can invert this to find k = VμL/AΔP.

If w is the “fracture length perpendicular to the flow direction”, as stated on lines 135-136, and dm is the mean aperture, then A = wdm. Hence, k = μV/(ΔP/L)wdm. But the mean aperture has been omitted from eq. (5). The authors then obtain the correct units, m2, by taking the 2/3rd power of the parenthesized term in eq. (5). But this step has no justification, as far as I can tell.

As dm is difficult if not impossible to measure in the field, it is more convenient to quantify the ability of the fracture to transmit fluids using the transmissivity, which is defined at T = kdm; see, for example, the paper “Hydraulic conductivity of rock fractures”, R. W. Zimmerman and G. S. Bodvarsson, Transport in Porous Media, 1996;23:1–30, which is reference [8] of the present paper. Using T has the effect of eliminating the need to know the value of dm. Perhaps it will turn out that correcting this error does not greatly change the main conclusions of this paper, but the present way of quantifying and presenting the results is not justifiable, in my opinion. 

2. An alternative explanation of why the permeability/transmissivity of a fracture falls below the value predicted by the naive “cubic law” has been given in the paper “A simple model for deviations from the cubic law for a fracture undergoing dilation or closure”, S. Sisavath, A. Al-Yaarubi, C. C. Pain, and R. W. Zimmerman, Pure Appl. Geophys., vol. 160, pp 1009-1022, 2003. The authors may want to mention and cite this alternative approach.

Round 2

Reviewer 2 Report

Comments and Suggestions for Authors

I have read the revised draft and the cover letter. I still do not agree with the author's definition of k in eq. (6). But as long as the authors have clearly explained their definition and their methodology, I do not think that readers will be confused or misled. So, I now recommend that the revised draft can be accepted for publication.